# Observer-based bipartite containment control of multi-agent systems with input and output quantization

Yanqing Hou
*Marine Electrical Engineering College*
*Dalian Maritime University*
Dalian, China
houyanqing0@163.com

Yan Yan
*Marine Electrical Engineering College*
*Dalian Maritime University*
Dalian, China
y.yan@dlmu.edu.cn

Shuanghe Yu
*Marine Electrical Engineering College*
*Dalian Maritime University*
Dalian, China
shuanghe@dlmu.edu.cn

*Abstract*—**This paper presents a sliding mode control (SMC) method for the bipartite containment control of multi-agent systems (MASs) under input and output quantization. First, an observer based on the super-twisting algorithm (STA) is used to estimate the unmeasured velocities of followers in finite time, using only quantized position information. Second, an SMC controller is developed to achieve bipartite containment control for followers under input and output quantization. The stability analysis ensures that sliding variables and bipartite containment position errors converge within specified ranges determined by the quantization parameters and STA-based observer parameters. Finally, simulation results validate the effectiveness of the proposed SMC method.**

*Index Terms*—**Sliding mode control (SMC), quantization, bipartite containment control, multi-agent system (MAS)**

## I. Introduction

In recent years, the cooperative control of multi-agent systems (MASs) has gained significant attention for tasks such as exploration of resources, marine geodesy and cartography, and unmanned aerial vehicle swarm show [1]. Studies have focused on fundamental MAS control mechanisms, including consensus, formation, and flocking strategies [2]. Among these, consensus has been a primary focus in cooperative control research [3]. Consensus problems are typically divided into three categories: leader-free consensus, single-leader trajectory tracking, and multi-leader containment control [4]. Containment control, where followers move into the convex hull formed by leaders, is particularly important for addressing practical issues [5].

Quantization refers to reducing data size by representing it with fewer bits or levels of precision, thereby conserving network bandwidth and minimizing communication overhead [17]. In practical applications, quantization control is widely employed in network information processing [7]. However, the discontinuity inherent in output quantization complicates controller design and stability analysis, receiving considerable attention [8].

This work was supported in part by the Natural Science Foundation of China under grant 62173054 and grant 62373072, and in part by the Natural Science Foundation of Liaoning under grant 2021-MS-142. (Corresponding author: Yan Yan.)

The super-twisting algorithm (STA) is extensively utilized in controllers and observers due to its continuous control signals and finite-time convergence properties [9] [10]. In [11], an STA-based observer is developed to estimate velocity and utilize actuator fault information to achieve multi-agent consensus despite actuator faults. Additionally, [12] introduces a third-order super-twisted extended state observer to estimate the position and velocity of a sensorless drive system in interior permanent magnet synchronous motors. Furthermore, [13] presents an adaptive STA-based observer with a predetermined convergence time in discrete time.

SMC is extensively applied across various applications due to its robust performance in the presence of external disturbances and uncertainties [14]. A periodic event-triggered SMC law is designed in [15] to solve the path-following problem of underactuated surface vehicles. The SMC is received even more attention under the influence of quantization [16]. Therefore, designing a quantized sliding mode controller in a MAS can enhance the robustness of the system while saving communication resources. This approach reduces the need for frequent updates of actuators, promoting energy efficiency and minimizing wear. Since quantized SMC of MASs is less researched, this paper investigates this direction.

The main contributions of this paper are summarized below.

1. In this paper, we address the bipartite containment control problem for MASs under both input and output quantization. We propose a sliding mode control (SMC) law specifically designed to tackle these challenges. The bipartite containment control strategy exhibits superior communication resource conservation compared to traditional strategies that only consider input quantization.

2. Given the availability of only quantized position signals, we design a super-twisting algorithm (STA)-based observer to use the unmeasured velocities of the followers within a finite time frame. The observation errors from this process are uniformly bounded and directly related to the chosen quantization parameters, ensuring reliable estimation performance.

3. Utilizing both input and output quantization, we develop a distributed control scheme that leverages the estimated

velocities. The stability of the MAS, despite external disturbances, parameter uncertainties, and the constraints of input and output quantization, is rigorously proven using the Lyapunov function. Additionally, we analytically determine the upper bounds of the bipartite containment position errors, demonstrating their dependence on the quantization parameters and the parameters of the STA-based observer.

*Notations:* $R$, $R^n$, and $R^{n \times n}$ denote the sets of all real numbers, $n \times 1$ column vectors, and $n \times n$ real matrices, respectively. $(\cdot)_{ij}$ represents the element of a matrix in row $i$ and column $j$. For a matrix $A$, $A^T$ and $A^{-1}$ denote its transpose and inverse, respectively. For $x \in R^n$, $\|x\|$ and $\|x\|_\infty$ denote the Euclidean norm and the infinity norm of $x$, respectively.

## II. PRELIMINARIES AND PROBLEM FORMULATION

### A. Algebraic Graph Theory

This paper discusses MASs consisting of $M$ leaders and $N$ followers. The communication network topology is represented by a directed graph $G = \{\mathcal{V}, \mathcal{E}\}$, where $\mathcal{V} = \{\nu_1, \nu_2, \ldots, \nu_{N+M}\}$ denotes the set of nodes, and $\mathcal{E} \subseteq \{\mathcal{V} \times \mathcal{V}\}$ denotes the set of edges. $(\nu_i, \nu_j) \in \mathcal{E}$ indicates that the information of the $j$th agent can be used for the $i$th agent and the opposite is false. Self-edges are not allowed in the graph, i.e., $(\nu_i, \nu_i) \notin \mathcal{E}$. In a signed graph, the nodes set consists of two subgroups $\mathcal{V}_1$ and $\mathcal{V}_2$, where $\mathcal{V}_1 \cup \mathcal{V}_2 = \mathcal{V}$ and $\mathcal{V}_1 \cap \mathcal{V}_2 = \emptyset$. Consider the adjacency matrix $A = [a_{ij}] \in R^{(N+M) \times (N+M)}$ of a directed graph $G = \{\mathcal{V}, \mathcal{E}\}$, if there exists an edge $(\nu_i, \nu_j) \in \mathcal{E}$, then $a_{ij} = 1$ $(\nu_i, \nu_j \in \mathcal{V}_q, q = 1, 2)$, which denotes the cooperative interaction, and $a_{ij} = -1$ $(\nu_i \in \mathcal{V}_p, \nu_j \in \mathcal{V}_q, p, q = 1, 2, p \neq q)$, which denotes the antagonistic interaction, and $a_{ij} = 0$ otherwise. Then the in-degree matrix $D$ is rewritten as $D = \text{diag}\{d_1, d_2, \ldots, d_{N+M}\}$, where $d_i = \sum_{j=1}^{N+M} |a_{ij}|$, and the Laplace matrix L is rewritten as $L = D - A$ with $l_{ii} = d_i$ and $l_{ij} = -a_{ij}$ for $i \neq j$. By using (1), the Laplace matrix is normalized to obtain $\ell$.

By normalizing the Laplace matrix L of the directed graph $G = \{\mathcal{V}, \mathcal{E}\}$, we get

$$\ell_{ij} = \begin{cases} 1, & \nu_i = \nu_j \\ \dfrac{l_{ij}}{l_{ii}}, & \nu_j \in \mathcal{L}_i \\ 0, & \text{otherwise} \end{cases} \tag{1}$$

Then the normalized directed Laplacian matrix $\ell$ can be rewritten as

$$\ell = \begin{bmatrix} \ell_1 & \ell_2 \\ 0_{M \times N} & 0_{M \times M} \end{bmatrix} \tag{2}$$

where $\ell_1 \in R^{N \times N}$, $\ell_2 \in R^{N \times M}$, and satisfied $-\ell_1^{-1} \ell_2 1_{M \times 1} = 1_{N \times 1}$.

### B. Quantizer

For saving communication resources, we consider the communication signals among leader and followers are quantized. We use the uniform quantizer which is defined as

$$q(\zeta) = \mu \, \text{round}\left(\frac{\zeta}{\mu}\right) \tag{3}$$

where $\zeta \in R$ is the input of the quantizer, $q(\zeta)$ is the quantized value and $\mu > 0$ is the quantization parameter. Define quantization error as $e_\zeta = q(\zeta) - \zeta$. It follows that $|e_\zeta| \leq 0.5\mu$.

### C. Problem Formulation

The dynamics of the $i$th follower are described by ($i = 1, \ldots, N$):

$$\begin{cases} \dot{x}_i = v_i, \\ \dot{v}_i = u_i + f(x_i, v_i, t) + d_i \end{cases} \tag{4}$$

where $x_i \in R^n$, $v_i \in R^n$ represent position and velocity of the $i$th follower, respectively, $u_i \in R^n$ is control input of the $i$th follower. And $f(x_i, v_i, t)$ is a nonlinear function, $d_i$ is an external disturbance.

**Definition 1.** Let there exist a set $C_h \subseteq R^n$. For any $\bar{x}, \bar{y} \in C_h$, and $\theta \in [0, 1]$, $C_h$ is said to be a convex set if the point $\theta \bar{x} + (1 - \theta \bar{y})$ is in $C_h$. Given the convex hull $Co(\chi)$ which is the minimal convex set containing all points in $\chi$ using a set of points $\chi = \{\chi_1, \chi_2, \ldots, \chi_N\}$, and is defined as $Co(\chi) = \{\sum_{i=1}^{N} \theta_i \chi_i | \chi_i \in \chi, \theta_i > 0, \sum_{i=1}^{N} \theta_i = 1\}$.

**Assumption 1.** Given a sequence of positive constants $\kappa_1, \kappa_2, \ldots, \kappa_M$ satisfying $\sum_{i=1}^{M} \kappa_i = 1$, for any $y$, $z$, $y_i$, $z_i \in R^n$, $i = 1, 2, \ldots M$ there exist two constants $\gamma_1, \gamma_2 > 0$ satisfying

$$\|f(y, z, t) - \sum_{i=1}^{M} \kappa_i f(y_i, z_i, t)\|_\infty$$
$$\leq \gamma_1 \|y - \sum_{i=1}^{M} \kappa_i y_i\|_\infty + \gamma_2 \|z - \sum_{i=1}^{M} \kappa_i z_i\|_\infty \tag{5}$$

The control objective of this paper is to design STA-based state observers and distributed bipartite containment controllers for MASs in the presence of unmeasured velocities and input and output quantization to make the position of each follower satisfy

$$\begin{cases} \lim_{t \to \infty} \|\eta_{1i} - h(t)\| \leq \delta, i \in \nu_1 \\ \lim_{t \to \infty} \|\eta_{1i} + h(t)\| \leq \delta, i \in \nu_2 \end{cases} \tag{6}$$

where $h(t) \in Co\{\eta_{1(N+1)}, \ldots, \eta_{1(N+M)}\}$ denotes the convex hull formed by leaders, and $\delta > 0$ is a constant.

## III. BIPARTITE CONTAINMENT CONTROL WITH OUTPUT AND INPUT QUANTIZATION

Define the error functions as

$$
\begin{cases}
e_{xi} = \sum_{j=1}^{N+M} a_{ij} \left( sign\left(a_{ij}\right) x_i - x_j \right) \\
e_{vi} = \sum_{j=1}^{N+M} a_{ij} \left( sign\left(a_{ij}\right) v_i - v_j \right)
\end{cases}
\quad i = 1, 2, \ldots, N \quad (7)
$$

We can get $\dot{e}_{xi} = e_{vi}$, and according to (7), we obtain

$$
\dot{e}_{vi} = \sum_{j=1}^{N+M} |a_{ij}| \left( f\left(x_i, v_i, t\right) + u_i\left(t\right) \right)
$$
$$
+ g_i(t) - \sum_{j=1}^{N} a_{ij} \left( f\left(x_j, v_j, t\right) + u_j\left(t\right) \right) - \sum_{j=N+1}^{N+M} a_{ij} \ddot{x}_j
$$
$$(8)$$

where $g_i(t) = \sum_{j=1}^{N+M} |a_{ij}| d_i(t) - \sum_{j=1}^{N} a_{ij} d_j(t)$ with $g_i(t) \leq 2d_{max}$.

The sliding mode variable $s_i$ is designed as

$$
s_i = c e_{xi} + e_{vi} \quad (9)
$$

From (8)-(9), the derivative of $s_i$ becomes

$$
\dot{s}_i = c e_{vi} + f\left(x_i, v_i, t\right) + u_i\left(t\right) + g_i(t)
$$
$$
- \sum_{j=1}^{N} a_{ij} \left( f\left(x_j, v_j, t\right) + u_j\left(t\right) \right) - \sum_{j=N+1}^{N+M} a_{ij} \ddot{x}_j \quad (10)
$$

By using the SMC technology and from (10), the control input of the $i$th follower can be taken as

$$
u_i(t) = - c e_{vi} - f\left(x_i, v_i, t\right) - K sign(s_i)
$$
$$
+ \sum_{j=1}^{N} a_{ij} \left( f\left(x_j, v_j, t\right) + u_j\left(t\right) \right) + \sum_{j=N+1}^{N+M} a_{ij} \ddot{x}_j
$$
$$(11)$$

where $K > 2d_{max}$.

Substituting the quantized control input into (3), we obtain

$$
\begin{cases}
\dot{x}_i = v_i, \\
\dot{v}_i = \hat{u}_i + f\left(x_i, v_i, t\right) + d_i
\end{cases}
\quad (12)
$$

where $\hat{u}_i(t) = q\left(u_i(t)\right)$. Let the quantization error of control inputs be $e_{ui}(t) = \hat{u}_i(t) - u_i(t)$, which satisfies $\|e_{ui}\|_\infty \leq 0.5\mu$.

In practice, the agent's actual position can be accurately obtained by GPS, but the velocity cannot [17]. Therefore, this paper considers a situation where the velocity is not measurable. Then, the STA-based observer is used to observe the velocity. The input signal of the STA-based observer is $\hat{x}_{ik}$, $i = 1, \ldots, N$, $k = 1, \ldots, n$. Let $\bar{x}_{ik}$ and $\bar{v}_{ik}$ be the estimates of signals $x_{ik}$ and $v_{ik}$, respectively. The STA-based observer equation is described as follows

$$
\begin{cases}
\dot{\bar{x}}_{ik} = \bar{v}_{ik} - k_1 |\bar{x}_{ik} - \hat{x}_{ik}|^{\frac{1}{2}} \text{sign}(\bar{x}_{ik} - \hat{x}_{ik}) \\
\dot{\bar{v}}_{ik} = f_{1ik}(\bar{x}_i, \bar{v}_i, t) - k_2 \text{sign}(\bar{x}_{ik} - \hat{x}_{ik})
\end{cases}
\quad (13)
$$

where $f_{1ik}$ is the $k$th element of $f_{1i}$, $k_1$ and $k_2$ are the observer parameters, and $f_{1i}(\bar{x}_i, \bar{v}_i, t) = f\left(\bar{x}_i, \bar{v}_i, t\right) + \hat{u}_i$.

Define the observation error variables $\tilde{x}_{ik} = \bar{x}_{ik} - x_{ik}$ and $\tilde{v}_{ik} = \bar{v}_{ik} - v_{ik}$. According to (12)-(13), the observation error equation is described as follows

$$
\begin{cases}
\dot{\tilde{x}}_{ik} = \tilde{v}_{ik} - k_1 |\bar{x}_{ik} - \hat{x}_{ik}|^{\frac{1}{2}} \text{sign}(\bar{x}_{ik} - \hat{x}_{ik}) \\
\dot{\tilde{v}}_{ik} = F_{1ik}(x_i, \bar{x}_i, v_i, \bar{v}_i) - k_2 \text{sign}(\bar{x}_{ik} - \hat{x}_{ik})
\end{cases}
\quad (14)
$$

where $F_{1ik}(x_i, \bar{x}_i, v_i, \bar{v}_i)$ is the $k$th element of $F_{1i}(x_i, \bar{x}_i, v_i, \bar{v}_i)$ and $F_{ik}(x_i, \bar{x}_i, v_i, \bar{v}_i) = f_{1ik}(\bar{x}_i, \bar{v}_i, t) - f_{1i}(x_i, v_i, t) - \hat{u}_i - d_i(t)$ is the aggregate uncertainty of the system.

**Assumption 2.** [18] The aggregate uncertainty term $F_{1i}$ is bounded and the upper bound is known, i.e. there exists a known positive constant $\bar{L}$ such that $|F_{1ik}(x_i, \bar{x}_i, v_i, \bar{v}_i)| \leq \bar{L}$.

**Theorem 1.** Consider the MAS (12) and the STA-based observer (13) with only uniformly quantized values of $x_{ik}$ are available. Under Assumption 2, if the observer parameters satisfy the conditions $k_1 > \mu$ and $k_2 > 3\bar{L} + \frac{2\bar{L}^2}{k_1^2}$, the observation error converges in finite time to

$$
\|\zeta_i\| \leq \chi_i \quad (15)
$$

where $\zeta_i = [\zeta_{1i}, \zeta_{2i}]^T = \left[ |\tilde{x}_{ik}|^{1/2} \text{sign}\left(\tilde{x}_{ik}\right), \tilde{v}_{ik} \right]^T$, $\chi_i = \sqrt{\frac{\lambda_{\max}\{P\}}{\lambda_{\min}\{P\}}} \Xi$, $\Xi = \frac{(k_1 \theta_1 + 2k_2 \theta_2)\sqrt{2\mu}}{(1-\bar{\kappa})\lambda_{\min}\{Q\}}$, $\theta_1 = \sqrt{p_{11}^2 + p_{12}^2}$, $\theta_2 = \sqrt{p_{12}^2 + 1}$, and $\bar{\kappa} \in (0,1)$ is a constant. Matrices $Q = \begin{bmatrix} q_{11} & q_{12} \\ q_{12} & q_{22} \end{bmatrix}$ and $P = \begin{bmatrix} p_{11} & p_{12} \\ p_{12} & p_{22} \end{bmatrix}$ are symmetric matrices with $q_{11} = k_1 \left(2k_2 + k_1^2 - 2\bar{L}\right)/2$, $q_{12} = -k_1 \left(k_1 + 2\bar{L}/k_1\right)/2$, $q_{22} = k_1/2$, $p_{11} = 2k_2 + k_1^2/2$, $p_{12} = -k_1/2$, and $p_{22} = 1$. And settling time of convergence $T_{si}$ is

$$
T_{si} = \frac{2\lambda_{max}^{\frac{1}{2}}}{\bar{\kappa}\lambda_{min}\{Q\}} \left[ V_{1i}^{\frac{1}{2}}(\tilde{\eta}_i(0)) - \lambda_{max}^{\frac{1}{2}}\{P\}\Xi \right] \quad (16)
$$

Detailed proof can be found in reference [10].

With only the quantized information being available, the tracking error $e_{xi}$ and $e_{vi}$ as defined in (7) become

$$
\begin{cases}
\hat{e}_{xi} = \sum_{j=1}^{N+M} a_{ij} \left( sign\left(a_{ij}\right) \hat{x}_i - \hat{x}_j \right) \\
\bar{e}_{vi} = \sum_{j=1}^{N+M} a_{ij} \left( sign\left(a_{ij}\right) \bar{v}_i - \bar{v}_j \right)
\end{cases}
\quad i = 1, 2, \ldots, N \quad (17)
$$

Due to quantization, the control input of the $i$th follower as defined in (11) become

$$
u_i(t) = - c\bar{e}_{vi} - f\left(\hat{x}_i, \bar{v}_i, t\right) - K_i sign(\check{s}_i)
$$
$$
+ \sum_{j=1}^{N} a_{ij} \left( f\left(\hat{x}_j, \bar{v}_j, t\right) + \hat{u}_j(t) \right) + \sum_{j=N+1}^{N+M} a_{ij} \hat{\ddot{x}}_j
$$
$$(18)$$

where

$$\check{s}_i(t) = c\hat{e}_{xi} + \bar{e}_{vi}. \tag{19}$$

We present the analysis to show the sliding variable $s_i$ can converge to a bounded region in finite time.

**Theorem 2.** Consider the MAS (12) with the sliding variable (19) and the SMC strategy (18) by using the quantized outputs and inputs information. If control parameters satisfy

$$K_i = 2(c + \gamma_2)\chi_i + (1 + \gamma_1)\mu + 2d_{max} + \varepsilon \tag{20}$$

where $\varepsilon > 0$ is a small constant, then the sliding variable $s_i$ can converge to the region $\|s_i\|_\infty \le c\mu + 2\chi_i$ and the overall bipartite containment position errors of each follower can converge to the region

$$\begin{cases} \lim_{t\to\infty} \|\eta_{1i} - h(t)\| \le \frac{\sqrt{n}(c\mu + 2\chi_i)}{c}, i \in \nu_1 \\ \lim_{t\to\infty} \|\eta_{1i} + h(t)\| \le \frac{\sqrt{n}(c\mu + 2\chi_i)}{c}, i \in \nu_2 \end{cases} \tag{21}$$

where $i = 1, 2, \ldots, N$.

*Proof:* Define the Lyapunov function as $V_i = \frac{1}{2} s_i^T s_i$. Considering $R(\psi_i) \approx R(\hat{\psi}_i)$ and substituting (18) into (10), we have

$$\begin{aligned} \dot{V}_i =& s_i^T(t)\dot{s}_i(t) \\ =& s_i^T(t)\bigg( f(x_i, v_i, t) - f(\hat{x}_i, \bar{v}_i, t) + e_{ui} + ce_{vi} - c\bar{e}_{vi} \\ & + \sum_{j=1}^N a_{ij}(f(\hat{x}_j, \bar{v}_j, t) - f(x_j, v_j, t)) \\ & + \sum_{j=N+1}^{N+M} a_{ij}\left(\ddot{\hat{x}}_j - \ddot{x}_j\right) + \sum_{j=1}^N a_{ij}(\hat{u}_j(t) - \hat{u}_j(t)) \\ & - K_i sign(\check{s}_i) + g_i(t) \bigg). \end{aligned} \tag{22}$$

From (3), we have

$$\begin{aligned} & \left\| \hat{e}_{xi}(t) - e_{xi}(t) \right\|_\infty \\ =& \left\| \sum_{j=1}^{N+M} a_{ij}(\hat{x}_i - x_i) - \sum_{j=1}^{N+M} a_{ij}(\hat{x}_j - x_j) \right\|_\infty \\ =& \left\| e_{1i} - \sum_{j=1}^{N+M} a_{ij} e_{1j} \right\|_\infty. \end{aligned} \tag{23}$$

From $\|e_{1i}\|_\infty \le 0.5\mu$, $\|e_{1j}\|_\infty \le 0.5\mu$, and $\|\sum_{j=1}^{N+M} a_{ij} e_{1j}\|_\infty \le 0.5\mu$, we have $\|\hat{e}_{xi}(t) - e_{xi}(t)\|_\infty \le \mu$. Similarly, we get $\|\bar{e}_{yi}(t) - e_{yi}(t)\|_\infty \le 2\chi_i$. From Assumption 1, it follows that

$$\begin{aligned} & \|f(x_i, v_i, t) - f(\hat{x}_i, \bar{v}_i, t)\|_\infty \\ \le& \gamma_1 \|x_i - \hat{x}_i\|_\infty + \gamma_2 \|v_i - \bar{v}_i\|_\infty \\ \le& \frac{\gamma_1 \mu}{2} + \gamma_2 \chi_i. \end{aligned} \tag{24}$$

After applying (23)-(24) to (22) yields

$$\begin{aligned} \dot{V}_i \le& \sum_{k=1}^n |s_{ik}|(2(c + \gamma_2)\chi_i + (1 + \gamma_1)\mu) \\ & + s_i^T(-K sign(\check{s}_i) + g_i(t)) \end{aligned} \tag{25}$$

Noting the term $\text{sgn}(\check{s}_i(t)) \in R^3$ in (25), we now establish the set $\Upsilon_{ik}$ where it is possible $\text{sgn}(\check{s}_{ik}(t)) \ne \text{sgn}(s_{ik}(t))$, $k = 1, \ldots, n$.

The difference between the sliding variable $\hat{s}_i$, which is affected by quantization, and the sliding variable $s_i$ staisfy

$$\|\check{s}_i - s_i\|_\infty = \|c(\hat{e}_{xi} - e_{xi}) + (\bar{e}_{vi} - e_{vi})\|_\infty \le c\mu + 2\chi_i \tag{26}$$

For $(e_{xi}, e_{vi}) \notin \Upsilon_{ik}$, $k = 1, 2, 3$, we have $\text{sgn}(\check{s}_i) = \text{sgn}(s_i)$. According to the condition (20), (25) becomes

$$\begin{aligned} \dot{V}_i \le& \sum_{k=1}^n |s_{ik}|(2(c + \gamma_2)\chi_i + (1 + \gamma_1)\mu) \\ & + s_i^T(-K sign(\check{s}_i) + g_i(t)) \\ \le& -\varepsilon \sum_{k=1}^n |s_{ik}| \\ \le& -\sqrt{2}\varepsilon V_i^{\frac{1}{2}} \end{aligned} \tag{27}$$

The settling time can be calculated as

$$T_i = T_{si} + \frac{\sqrt{2} V_i^{\frac{1}{2}}(T_{si})}{\varepsilon}, i = 1, 2, \ldots, N \tag{28}$$

which means that the sliding variable $s_i$ of the $i$th follower must enter the quasi-sliding mode bound when $t \ge T_i$.

Next, we show the actual position of followers and the farthest distance at corresponding the convex hull. According to (9) and (26), we obtain

$$\begin{aligned} \|e_{xi}(t)\| =& \|e_{xi}(0)e^{-ct} + \int_0^t e^{-c(t-\tau)}s(\tau)d\tau\| \\ <& \|e_{xi}(0)\|e^{-ct} + \sqrt{n}(c\mu + 2\chi_i)\int_0^t e^{-c(t-\tau)}d\tau \\ =& \|e_{xi}(0)\|e^{-ct} + \frac{\sqrt{n}(c\mu + 2\chi_i)}{c}(1 - e^{-ct}). \end{aligned} \tag{29}$$

Since $c > 0$, we have $\|e_{xi}(t)\| < \frac{\sqrt{n}(c\mu + 2\chi_i)}{c}$ as $t \to \infty$. Therefore, the distance between the position of the $i$th follower and ideal position should satisfy

$$\lim_{t\to\infty} \|\eta_{1i} - h(t)\| \le \frac{\sqrt{n}(c\mu + 2\chi_i)}{c}$$

This ends the proof. ∎

## IV. SIMULATION RESULTS

In this section, we give a simulation to demonstrate the bipartite containment control for MASs. Consider the MAS consisting of three leader agents and four followers. The trajectories of each leaders are $x_5 = [0.1t, 2.5\sin(\frac{t}{30}) + 10, \arctan(\frac{5}{6}\cos(\frac{t}{30}))]^T$, $x_6 = [0.1t, 2.5\sin(\frac{t}{30}) - 10, \arctan(\frac{5}{6}\cos(\frac{t}{30}))]^T$, and $x_7 = [0.1t + 3, 2.5\sin(\frac{t}{30} + 1) - 2, \arctan(\frac{5}{6}\cos(\frac{t}{30} + 1))]^T$. The initial positions of the followers are given as $x_1 = [-5.625, 9.325, 0]^T$, $x_2 = [-3.75, -3.75, \frac{3\pi}{4}]^T$, $x_3 = [0, 3.75, \frac{3\pi}{8}]^T$, and $_4 = [-3.75, -9.375, \frac{3\pi}{8}]^T$. The initial velocities of all followers are $[0, 0, 0]^T$. The external disturbance to the followers is $d_i(t) = [\sin(0.2t + \frac{\pi}{5}) + 2\text{rand}() - 1, \cos(t - \frac{\pi}{6}) + 2\text{rand}() - 1, 0.1\cos(t) + 2\text{rand}() - 1]^T$, where $\text{rand}()$ is a random number

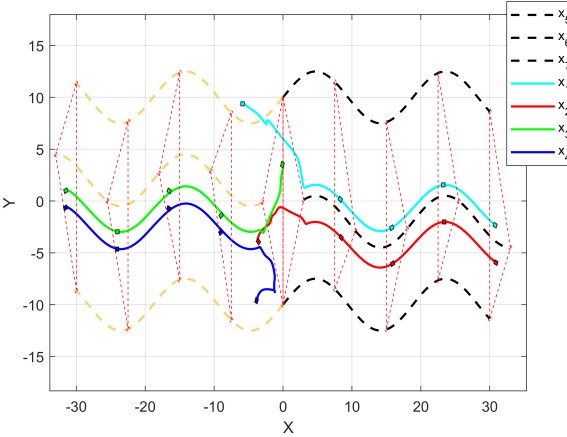

Fig. 1. Trajectories of all agents.

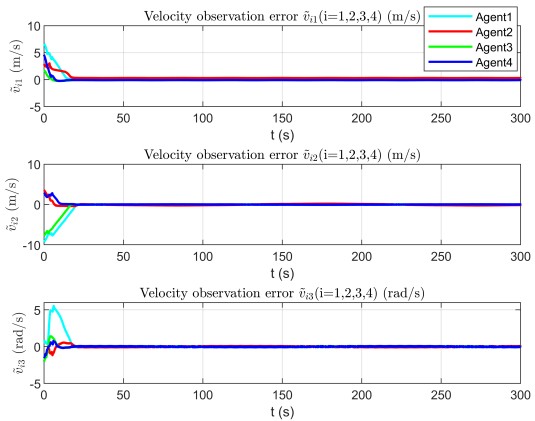

Fig. 2. Velocity observation errors of followers.

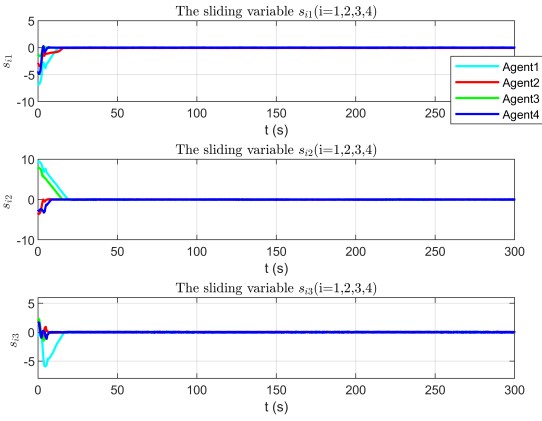

Fig. 3. The sliding variables of the followers.

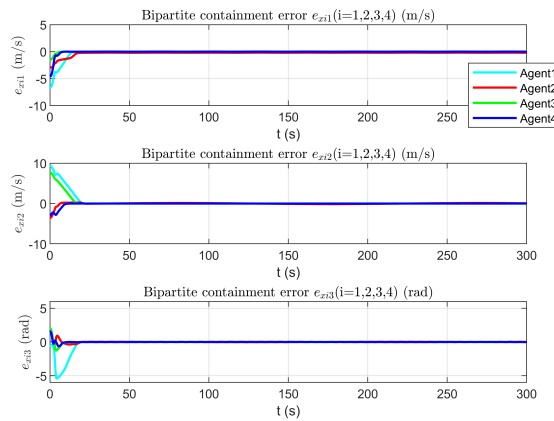

Fig. 4. Bipartite containment errors of followers.

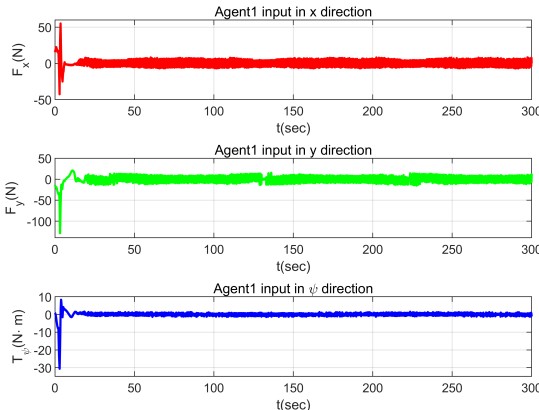

Fig. 5. Control inputs of the first follower agent.

Fig. 1 shows the trajectories of all agents during the interval from 0 to 300 seconds. The figure highlights four specific moments: $t = 75\mathrm{s}, 150\mathrm{s}, 225\mathrm{s}$, and $300\mathrm{s}$. At these times, the follower agents are seen entering the convex hull formed by the leaders and the symmetric convex hull. Fig. 2 indicates that the velocity observation errors for the STA-based observers can converge to a bounded region in a finite time, as given in Theorem 1. Fig. 3 illustrates the sliding variables $s_i$ of the four follower agents. It is evident that the sliding variables $s_{ik}$ with $k = 1, \ldots, n$ converge to a bound. The bounds $\Upsilon_{ik}$ and the settling time of the sliding variables are provided in Theorem 2. Fig. 4 presents the containment position errors of three follower agents under input and output quantization. To save space, only the control inputs of the first follower agent are shown in Fig. 5.

## V. CONCLUSION

In this paper, we studied sliding mode bipartite containment control for MASs under input and output quantization. We addressed the challenge of unmeasurable follower velocities by designing an STA-based observer to estimate these velocities

ranging from zero to one. The parameters of the sliding mode controllers are selected as $c = 1$ and $\chi = 0.1$. The quantization parameter is $\mu = 0.01$.

within a finite time. Using quantized position and observed velocity information, we developed a distributed bipartite containment control law that enables the followers to converge near the convex hull formed by the leaders and the symmetric convex hull. Our analysis shows that the sliding variable and bipartite containment position errors converge to a bounded region determined by the quantization parameters and the upper bound of observation errors. Simulation results support our theoretical findings.

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
