# OpenReview forum: "Observer-based bipartite containment control of multi-agent systems with input and output quantization"
_IEEE.org/ICIST/2024/Conference — IEEE ICIST 2024 Conference Submission_

### Official Review · Reviewer_uAMZ · 2024-08-26
**Accept**

**Rating:** 10
**Confidence:** 5

**Review:**

This paper presents a sliding mode control (SMC) method for the bipartite containment control of multi-agent systems (MASs) under input and output quantization. Overall, the language and organization are satisfactory. It can be accepted as a conference paper.

---

### Official Review · Reviewer_WjCp · 2024-08-29
**Accept after modification**

**Rating:** 7
**Confidence:** 4

**Review:**

This paper primarily proposes a sliding mode control approach for bipartite containment control in multi-agent systems with input and output quantization. Overall, the article presents a comprehensive narrative. However, several points merit attention:

1.The manuscript contains minor grammatical and formatting issues, such as the incorrect ordering of references (e.g., reference 17 appears before reference 7 in the introduction) and inaccuracies in the phrasing, such as the term ‘an STA-based observer’ following a reference. It is advisable to review the entire article to enhance its rigor and readability.
2.In the introduction, the acronym SMC should be introduced with its full form upon its first appearance.
3.The conclusion stating ‘It follows that |eζ| ≤ 0.5µ.’ lacks adequate justification within the text. Please provide a reference or additional explanation to substantiate this conclusion.
4.Figure 4 displays incorrect units for the variable x. The units for position are incorrectly listed as m/s, whereas they should be in meters (m).

---

### Official Review · Reviewer_XdTU · 2024-08-30
**This paper can be accepted.**

**Rating:** 7
**Confidence:** 3

**Review:**

1.	Innovation: The paper suggests a new control method using sliding mode control (SMC) and super-twisting algorithm (STA) for managing multi-agent systems (MAS) with quantized inputs and outputs. It’s an interesting approach, but we need more experiments and real-world tests to see if it really makes a difference compared to existing methods.

2.	Data Sources and Methods: The study relies on simulated data to test the proposed method. While the simulations support the theory, it’s missing comparisons with real-world data or more complex scenarios. It’d be good to discuss how this method might work in real life and clarify how the simulation data was created and why certain model parameters were chosen.

3.	Theory: The theoretical part is solid with clear stability analysis and control design. However, some explanations and formulas are a bit brief. Adding more details or an appendix for key steps would help readers follow the derivation better.

4.	Experiments: The experiments are well done and show the method’s effectiveness. However, they only cover a few scenarios. Testing in more complex situations, like with different disturbances and quantization levels, would strengthen the results.

5.	Figures and Tables: The figures and tables are clear, but some could use better color or line styles to improve readability. More detailed annotations could also help explain the results better.

6.	Comparisons: The paper compares the new method with existing sliding mode and quantization control methods, which is good. But it could also benefit from comparing with other advanced control techniques, like robust or fuzzy control, to provide a fuller picture of its strengths and weaknesses.

7.	Spelling and Grammar: The paper is mostly well-written with minor grammar issues. Some sentences are a bit wordy and could be simplified. A quick language check could improve clarity, especially for longer sentences.

8.	Conclusion: The conclusion wraps up the research well, but adding a look at future research directions or potential applications would make it even stronger and more forward-thinking.

---

### Decision · Program_Chairs · 2024-09-06

Accept (Oral)